# Pharmacological intervention of behavioural traits and brain histopathology of prenatal valproic acid-induced mouse model of autism

**Sharmind Neelotpol**[1☯]*, **Rifat Rezwan**[1☯], **Timothy Singh**[1], **Iffat Islam Mayesha**[1], **Sayedatus Saba**[2], **Mohd Raeed Jamiruddin**[1]

**1** School of Pharmacy, Brac University, Dhaka, Bangladesh, **2** Department of Clinical Pathology, Dhaka Medical College Hospital, Dhaka, Bangladesh

☯ These authors contributed equally to this work.
* sharmind@bracu.ac.bd

## Abstract

Autism spectrum disorder (ASD) is one of the leading causes of distorted social communication, impaired speech, hyperactivity, anxiety, and stereotyped repetitive behaviour. The aetiology of ASD is complex; therefore, multiple drugs have been suggested to manage the symptoms. Studies with histamine H3 receptor (H3R) blockers and acetylcholinesterase (AchE) blockers are considered potential therapeutic agents for the management of various cognitive impairments. Therefore, the aim of this study was to evaluate the neuro-behavioural effects of Betahistine, an H3R antagonist, and Donepezil, an acetylcholinesterase inhibitor on Swiss *albino* mouse model of autism. The mice were intraperitoneally injected with valproic acid (VPA) on the embryonic 12.5th day to induce autism-like symptoms in their offspring. This induced autism-like symptoms persists throughout the life. After administration of different experimental doses, various locomotor tests: Open Field, Hole-Board, Hole Cross and behavioural tests by Y-Maze Spontaneous Alternation and histopathology of brain were performed and compared with the control and negative control (NC1) groups of mice. The behavioural Y-Maze test exhibits significant improvement ($p < 0.01$) on the short term memory of the test subjects upon administration of lower dose of Betahistine along with MAO-B inhibitor Rasagiline once compared with the NC1 group (VPA-exposed mice). Furthermore, the tests showed significant reduction in locomotion in line crossing ($p < 0.05$), rearing ($p < 0.001$) of the Open Field Test, and the Hole Cross Test ($p < 0.01$) with administration of higher dose of Betahistine. Both of these effects were observed upon administration of acetylcholinesterase inhibitor, Donepezil. Brain-histopathology showed lower neuronal loss and degeneration in the treated groups of mice in comparison with the NC1 VPA-exposed mice. Administration of Betahistine and Rasagiline ameliorates symptoms like memory deficit and hyperactivity, proving their therapeutic effects. The effects found are dose dependent. The findings suggest that H3R might be a viable target for the treatment of ASD.

**Data Availability Statement:** All relevant data are within the manuscript and its Supporting Information files.

**Funding:** The author(s) received no specific funding for this work.

**Competing interests:** The authors have declared that no competing interests exist.

**Abbreviations:** ASD, Autism Spectrum Disorder; H3R, Histamine 3 Receptor; VPA, Valproic Acid; OFT, Open Field Test; HBT, Hole Board Test; HCT, Hole Cross Test.

## Introduction

Autism Spectrum Disorder (ASD) is an umbrella term that accounts for a number of neurodevelopmental conditions with a wide range of symptoms with varying severity. Adoption of repetitive behaviours, impairment of social interactions and relations are common traits of ASD [1]. ASDs begins in childhood and persist throughout life [2,3]. However, it cannot be detected prior to 2-3 years of maturity in a child as the symptoms are not identifiable in infants. The frequent episodes that are observed in individuals with ASD are epilepsy, depression, anxiety, and attention deficit hyperactivity disorder (ADHD) [4].

Recent studies showed that the median global autism prevalence rate is 100/10,000 (range: 1.09/10,000 to 436.0/10,000), where the median male-female ratio is 4:2. A median of 33.0% of autism cases with co-occurring intellectual disability was recorded [1]. Presently, the prevalence rate of autism is higher which maybe due to improved awareness, expansion of diagnostic criteria, better diagnostic tools and improved reporting.

Irrespective of an increase in the prevalence of ASD, the pathophysiological framework of ASD has not yet been fully established. A combination of heterogeneous factors such as genetic predisposition, and environmental exposure mainly during pregnancy are the known causative factors of autism [2]. Currently, there are no objective diagnostic test and a cure is yet to be found. Risperidone and aripiprazole has been approved by FDA for the control of ASD symptom [5]. However, no particular treatment holds promising therapeutic effects for all autistic individuals. Therefore, a successful pharmacological intervention strategy is required to improve symptomatic behaviours.

Recent studies showed that the brain histaminergic system is one of the attractive pharmacological targets for therapeutic purposes. The histaminergic system is involved in modulating cognition and behaviour and can play a role in microglial activation and neuroinflammation [6]. Henceforth, researchers focused on different histamine receptor (H1R, H2R, H3R, and H4R) antagonists to treat VPA-exposed mice [7–12]. Niaprazine, famotidine, scopolamine, which are H1R, H2R and, H3R antagonists, respectively, showed behavioural improvement in ASD [9], schizophrenia (SCH) [10], Alzheimer's disease (AD), and narcolepsy [13]. Studies showed that multitargeting histamine and dopamine receptors also yielded significant results in ameliorating neuronal oxidative stress and repetitive behaviours [14].

Histamine or its analogues through receptor binding mediates multiple brain functions in addition to homeostasis and immunity such as circadian-feeding cycle, sleep-wake regulation and learning [15]. Studies showed that antagonism of H3 hetero-receptors accelerates the corticolimbic liberation of acetylcholine, norepinephrine, glutamate, dopamine, serotonin, and gamma-aminobutyric acid (GABA) [7–12]. Such antagonism regulates higher brain function and maintains homeostasis of the nervous system [15]. Interestingly, Rasagiline (N-Propargyl-1-[R]-aminoindan), a selective, reversible MAO-B inhibitor, has been shown to increase levels of striatal extracellular dopamine at a lower dose (over 2 weeks) [16,17]. Similarly, a combination of H3 receptor blocker and MAO-B inhibitor shows a synergistic effect in the increase of dopamine in the brain [18].

The cholinergic system is responsible for various key functions such as memory formation, cognitive flexibility, and brain plasticity [19,20], besides it being a key mediator of early stage neural and synaptic development [21]. Decreased acetylcholine levels were observed in the temporal lobe and grey matter of ASD phenotypes which serves as an indicator that cholinergic imbalance might have a link with the pathophysiology of ASD. Furthermore, VPA-exposed mice have been previously reported to have a decreased choline peak level due to increased acetylcholinesterase up-regulation in the synaptic cleft [19]. Delayed and malformed cortical

neuronal development in cholinergic neurons disrupted early postnatal brain development in rats along with hypertrophy and hyperplasia [21].

Research on the histaminergic pathway as well as the cholinergic system could shed some light on the ASD pathophysiology and might bring viable treatment options in future as both of the pathways have been previously reported to be involved in development of ASD phenotypic outcomes [14,22]. Therefore, the aim of this study was to perform observational tests of varying doses of acetylcholinesterase inhibitor Donepezil, and histamine H3 inhibitor, Betahistine, both alone and in combination with MAO-B inhibitor Rasagiline, and its ameliorating effects on autism-like behaviour and brain neuronal degradation.

## Materials and methods

### Materials

**Chemicals.** Valproic acid sodium salt (P4543-10G) and sodium chloride were provided by Sanofi Aventis Bangladesh (Sigma-Aldrich Co., USA). Betahistine dihydrochloride (H3R antagonist), Rasagiline mesylate (MAO-B inhibitor), and Donepezil hydrochloride monohydrate (acetylcholinesterase inhibitor) were provided by Square Pharmaceuticals Limited, Bangladesh (Hetero Drugs Ltd., India).

**Animals.** Healthy male and female *Swiss albino* mice, 8 weeks of age, weighing 30–35 g were obtained from the Pharmacology Laboratory, Jahangirnagar University. Animals were housed in polypropylene cages with proper bedding, water and food. The environment was regulated at 25°C±1°C temperature and 45–55% relative humidity, with a 12:12 h light/dark cycle.

### Methods

**Preparing animals for the experiment.** Male and female mice were allowed to mate overnight. Pregnancy confirmation was carried out by the observation of vaginal plug or spermatozoa and was designated as embryonic day 0.5 (E0.5). The pregnant mice were caged separately and distributed randomly into two groups: VPA group (n=10), and saline group (n=5). Sodium valproate was dissolved in isotonic 0.9% sodium chloride solution and was given intraperitoneally on the embryonic day 12.5 (E12.5) [23,24] at a dose of 600 mg/kg [25,26]. On the other hand, another group of mice received the same volume of normal saline without sodium valproate at the same time (Control group). The day of birth of the offspring was considered as the postnatal day zero (P0). The mice pups were randomly caged and allowed to be weaned till postnatal day 21 (P21). After P21, the offspring of the same sex were housed separately (4-5 per cage).

**Preparation of drug doses.** Betahistine dihydrochloride, Rasagiline mesylate, and Donepezil hydrochloride monohydrate were administered upon mixing with water. An oral dose of Betahistine dihydrochloride of 3mg/kg was administered in combination with 0.8mg/kg of Rasagiline (T1) [21–23] and a larger dose of 30mg/kg was administered alone (T2) [21,23]. Three Donepezil oral doses of 0.3mg/kg (T3), 0.6mg/kg (T4) and 1mg/kg (T5) were administered [24]. All the reagents used in the study were of analytical grade. The doses were dissolved in water and given in 0.5ml solution.

**Study design.** All the animals were treated humanely throughout the experimental period and maximum care was taken in case of handling following the internationally accepted guide for the care and use of laboratory animals, published by the US National Institutes of Health (NIH Publication No. 85-23, Revised in 1985). The offspring were divided into 8 groups where each group consists of 6 mice. Thirty-six autistic offspring were taken from a VPA induced mother: five treated groups (T1, T2, T3, T4 and T5) and a negative control group 1 (NC1) (0.5

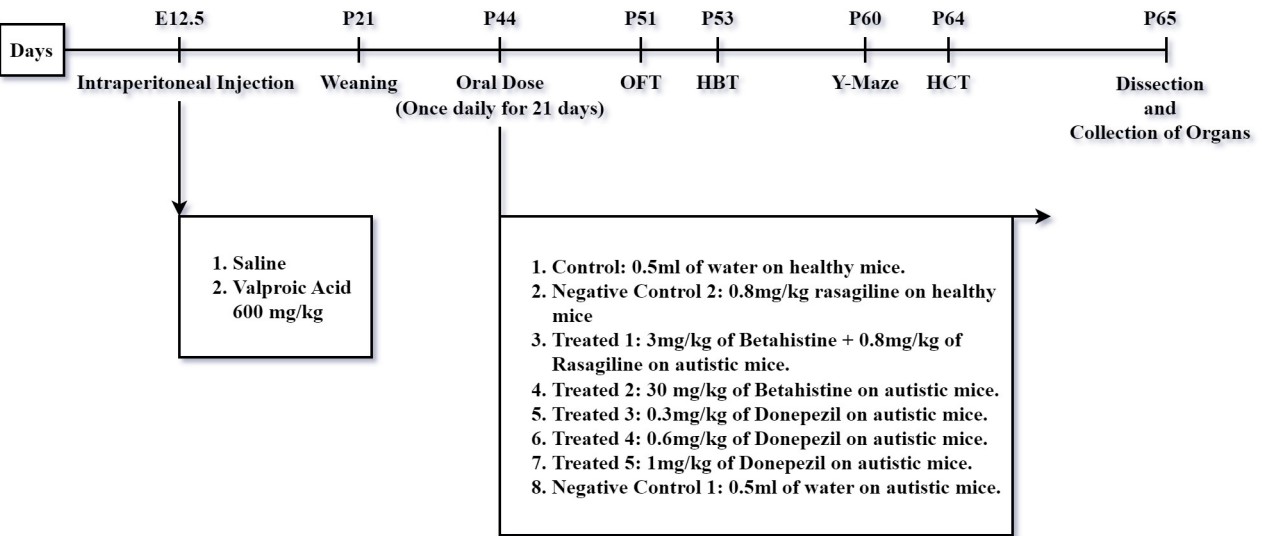

**Fig 1. Schematic diagram of doses, and time schedule for the behavioural and locomotor tests in healthy and VPA-induced mice.** Pregnant mice were given VPA (600 mg/kg, i.p.) on embryonic day 12.5 (E12.5). Drugs were given by oral gavage from postnatal day 44 (P44) until postnatal day 64 (P64) for a total of 21 days. Behavioural tests were conducted starting from P51. OFT: Open Field Test; HBT: Hole Board Test; HCT: Hole Cross Test.

ml of water) were created from them. Healthy offspring from the healthy mothers were divided into two groups – a control group (0.5 ml of water) and a negative control group 2 (0.8mg/kg Rasagiline). All the experimental mice were housed with ad libitum access to food and water and were caged in a (60 ×38 ×20 cm$^3$) cage under a controlled temperature of 21±5°C with a 12-h light/dark cycle. All the doses were administered once per day by oral gavage for 21 days from postnatal day 44 (P44) until postnatal day 64 (P64). Starting from postnatal day 51 (P51), the behavioural tests and locomotor tests were carried on. All the doses and saline were administered 30-40 minutes before each behavioural test. All the tests were conducted within 30 to 45 minutes after administration of the doses and between 9:00 am and 3:00 pm (Fig 1). After the completion of the neurobehavioural tests, each mouse was sacrificed following the study protocol and prepared for histopathological test.

**Behavioural tests.** *Y-maze test.* The Y-maze consisted of 3 limbs marked A, B and C. All the limbs of the maze were 120° apart from one another. The dimensions of each arm were (21 x 7 x 15.5 cm). The mouse was placed in the distal part of the arm of the instrument, facing the centre. Then it was allowed to roam inside the maze freely for 8 minutes. All of the arms were opened during this test. The session was recorded. After testing all the mice, the records were observed and their alternations of the arm entries were enumerated along with their total arm entry. Percentage (%) Alternation was calculated using the following formula.

$$\% \ Alternation = \frac{Number \ of \ Alternations}{[(Total \ number \ of \ arm \ entries - 2) \times 100]}$$

The maze was cleaned with a 70% ethanol solution after testing each subject [25,26].

**Locomotor tests.** *Open-field Test (OFT).* In this test, a square shaped box with dimensions of (50 × 50 × 35 cm) was used. The mice were placed in the centre of the open field box and were allowed to explore freely inside the box for 5 minutes. The experiment was conducted in a moderately lighted condition. The activities and movements of the mice were recorded [27].

The following parameters were monitored: 1) Line crossing: Frequency of mouse crossing grid lines with all four paws 2) Rearing: the mouse reared on its hind paws while in the

peripheral 3) Central area frequency: the frequency of mouse entry with all four paws in the arena centre 4) Defecation and urination [28]. Furthermore, the box was cleaned with a 70% ethanol solution after testing each subject.

*Hole-Board Test (HBT).* In this test, a wooden hole-board apparatus was used, with dimensions of ($68 \times 68 \times 40$ cm). The hole-board was made of wooden apparatus with 16 evenly spaced holes (3 cm in diameter) and 25 cm in height. A mouse was placed at the centre of the device and allowed to explore the apparatus for 5 minutes. The following behavioural patterns were observed and recorded: 1) new area entry/Line crossing: entry inside a new area with all of its four paws. 2) head-dip: the subject dips its head to a minimum depth of ear level so that the ear is levelled with the floor. The test was conducted in dimmed lighting conditions for 5 minutes [29]. After each trial the apparatus and objects were cleaned with 70% ethanol solution.

*Hole-Cross Test (HCT).* A wooden hole-cross apparatus was used for this test. The apparatus had a dimension of $30 \times 20 \times 14$ cm. A fixed partition with a 3 cm hole carved inside was placed in the middle. Each test mouse was allowed to roam in the apparatus without any disturbance for 5 minutes and the number of passages were recorded. The apparatus was cleaned with a 70% ethanol solution after testing each subject [30,31].

**Histopathology.** At the end of the 64th day of the experiment, the animals were euthanized as per the study protocol, using an overdose of ketamine (500 mg/kg, i.p.) and the anaesthetic effect was confirmed by "toe pinch". The whole brains of the mice were excised, weighed and preserved in 10% neutral buffered formalin. After few days, the brains of each mouse were routinely processed for histological studies using the Haematoxylin and Eosin (H&E) staining method. Finally, those tissues were examined under a light microscope (Olympus, CX43, Japan) for visualization. Photomicrographs of the brain sections were taken with the help of Olympus, D22 camera.

## Statistical analysis

Statistical analyses were carried out by using IBM SPSS Statistics version 26 for Windows (SPSS, Chicago, IL). The data were checked for skewness and the test for normality was performed using Q-Q plot. One-way analysis of variance (ANOVA) was performed in multiple pairwise comparisons followed by Fisher's least significant difference (LSD). *p*-values less than 0.05 were considered as statistically significant.

## Ethical approval

Ethical approval has been granted by the Biosafety, Biosecurity & Ethical Committee, Faculty of Biological Sciences, Jahangirnagar University, Savar, Dhaka, Bangladesh (Ref No: BBEC, JU/M 2020 (9)2). Throughout the experimental period animals were treated and handled as per the internationally accepted guideline for the care and use of laboratory animals, published by the US National Institutes of Health (NIH Publication No. 85-23, Revised in 1985).

## Result

### Y-maze test

In the Y-maze test, there was a significant difference of spontaneous alternation % between the control group (C) and the negative control 1 group (NC1) of mice ($p < 0.001$). In addition, the treated group 1 (T1: 3 mg/kg Betahistine + 0.8 mg/kg Rasagiline) ($p < 0.01$), treated group 2 (T2: 30 mg/kg of Betahistine) ($p < 0.05$) (Fig 2A), treated group 3 (T3: 0.3mg/kg of Donepezil) ($p < 0.001$), treated group 4 (T4: 0.6 mg/kg of Donepezil) ($p < 0.001$), and treated group 5

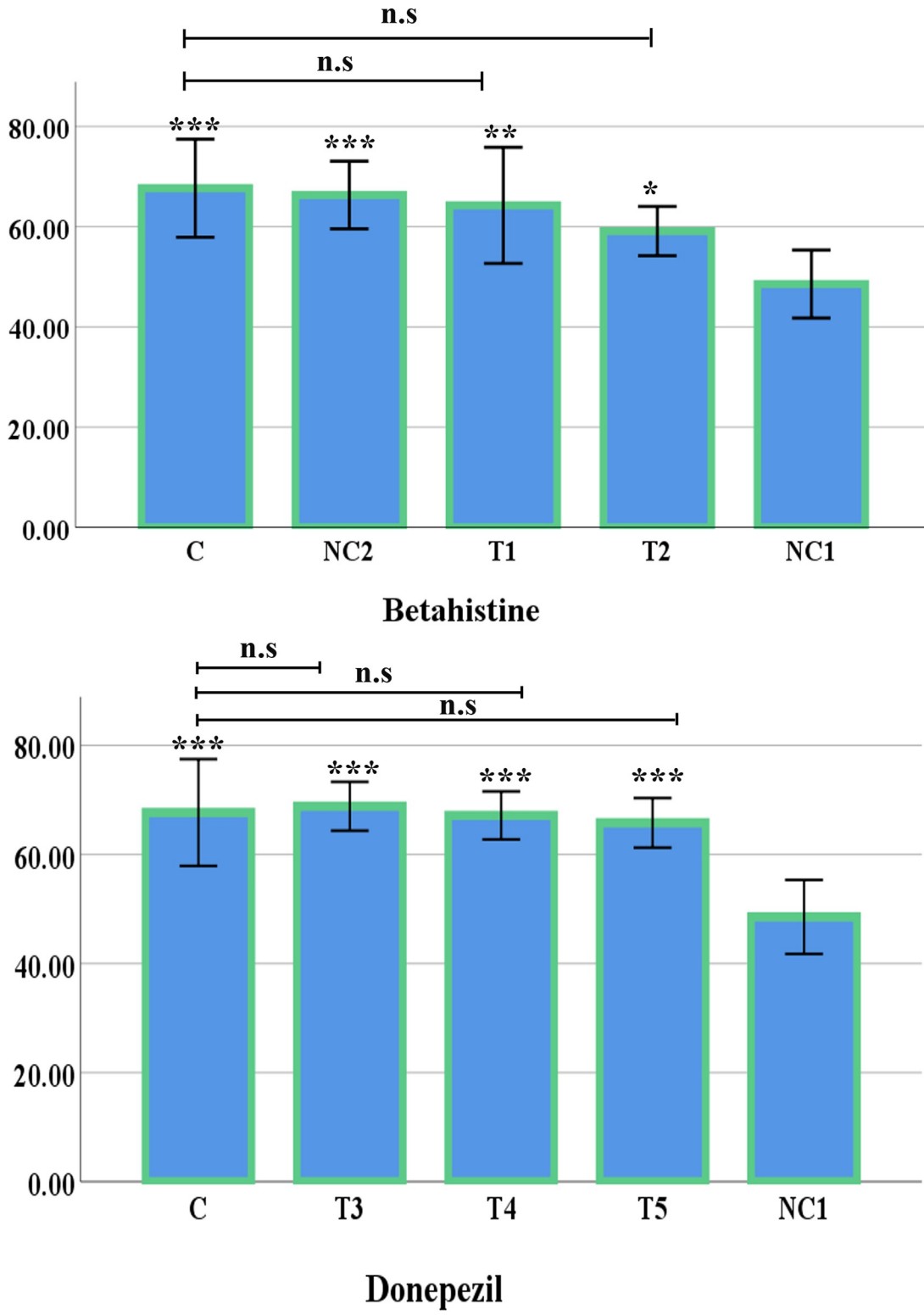

**Fig 2. Graphical representation of spontaneous alternation percentage in Y-maze test (Mean±SE).** (A) Spontaneous alternation percentage in Y-maze test of Betahistine, compared to negative control group 1. (B) Spontaneous alternation percentage in Y-maze test of Donepezil, compared to negative control group 1. (*$p < 0.05$, **$p < 0.01$ and ***$p < 0.001$). When compared with the control group, the treated groups did not show any significant difference (indicates as n.s i.e. non-significant).

(T5:1 mg/kg of Donepezil) ($p < 0.001$) (Fig 2B) demonstrated a significantly increased level of alternation compared to the negative control 1 group. Moreover, the changes found between the control group (healthy mice) and the treated groups were insignificant (Fig 2).

## Open-field test

In the open field test certain parameters were observed, such as: line crossing, rearing, central area frequency, urination and defecation (Fig 3).

Line Crossing: In the open-field line crossing test, there was a significant difference between control group (C) and negative control 1 group (NC1) ($p < 0.001$). In the Betahistine doses, the

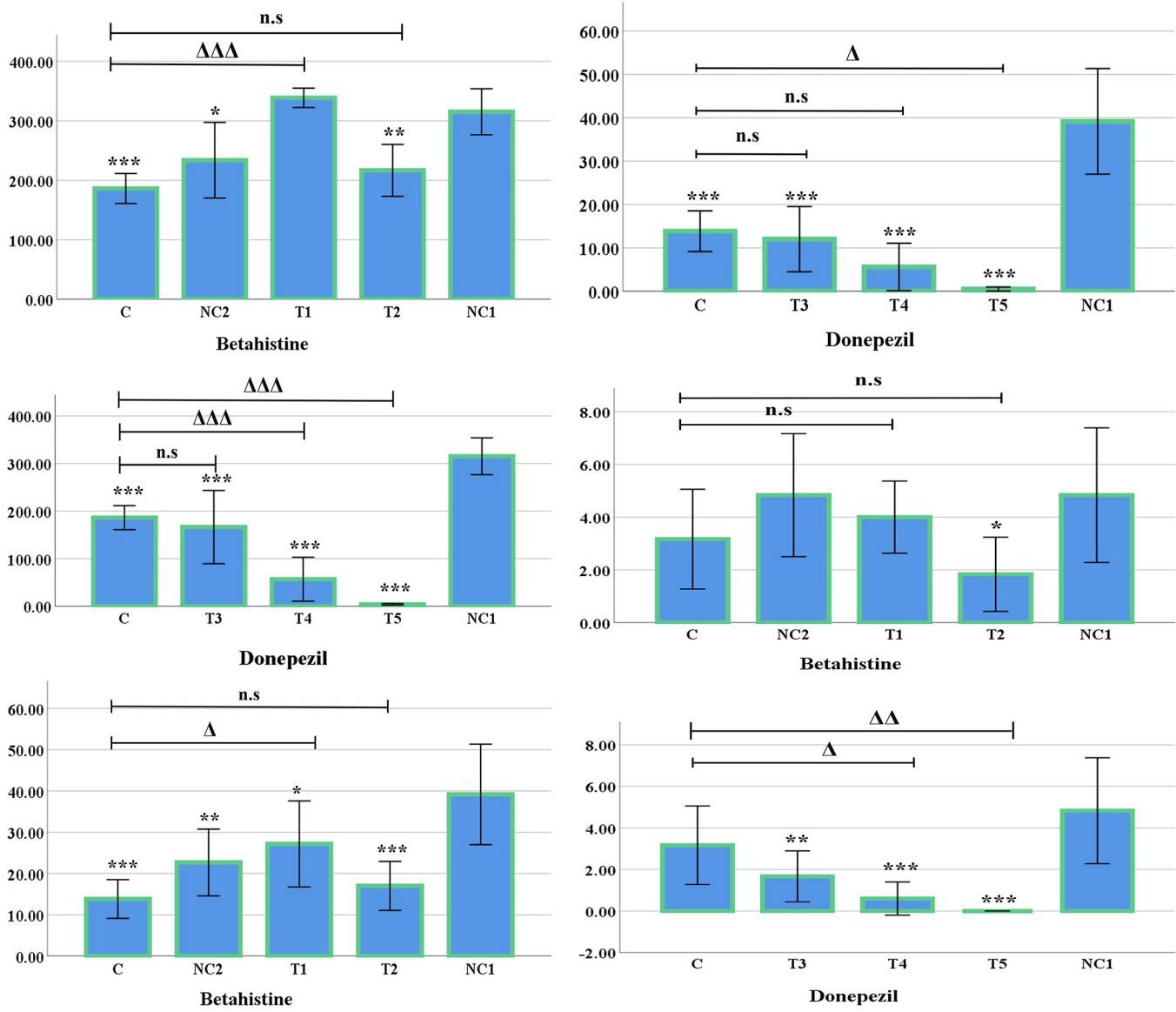

**Fig 3. Graphical representation of Open-field test (Mean±SE).** (A&B) Line crossing test of Betahistine and Donepezil, respectively. (C&D) Rearing test of Betahistine and Donepezil, respectively. (E&F) Central area frequency test of Betahistine and Donepezil, respectively. The following symbols: *, **, *** and Δ, ΔΔ, ΔΔΔ represents level of significance (p-values) at 5%, 1% and 0.1% level, when compared the treated groups with negative control 1 group and control group, respectively. n.s indicates non-significant.

T1 group did not show any significant difference over the negative control 1 group, while the T2 group demonstrated a significantly decreased line crossing ($p < 0.01$). However, no significant difference was observed between the control group and the negative control 2 group (0.8 mg/kg Rasagiline on healthy mice). In the case of Donepezil, the T3, T4, and T5 group all exhibited a significantly lower line crossing in comparison with the negative control 1 group ($p < 0.001$). Moreover, the T4 group and T5 group demonstrated a significant decrease in line crossing ($p < 0.001$) with the control group (Fig 3A and 3B).

In the case of the open-field rearing test, a significantly lower movement was observed between the Control group and the NC1 group ($p < 0.001$). The T1 group showed a significant decrease ($p < 0.05$), while the T2, T3, T4, and T5 group exhibited a highly significant decrease ($p < 0.001$) in rearing, compared to the negative control 1 group. Although, both the T1 and T5 groups showed a significant difference ($p < 0.05$) when compared to the control group, no significant difference was observed between the control group and the negative control 2 group in this regard (Fig 3C and 3D).

The negative control 1 group did not show any significant difference from the control group in the case of open-field central area frequency. While the T1 group did not show any significant difference, the T2 ($p < 0.05$), T3 ($p < 0.01$) and T4 & T5 group ($p < 0.001$) showed significantly decreased effect when compared with the negative control 1 group. Moreover, the T4 ($p < 0.05$) and the T5 group ($p < 0.01$) showed a significant reduction compared to the control group. In addition, the control group and negative control 2 group did not have any significant difference (Fig 3E and 3F).

While in urination the overall tests did not show any significant difference across the groups, there were significant differences in the frequency of defecation between the control and the negative control 1 group. The T3 group had significantly decreased ($p < 0.05$) defecation frequency in comparison with the NC1 group. On the contrary, the T1, T2, T4 and T5 group did not show any significant difference with the NC1 group. However, a significantly decreased effect was observed between the T3 group ($p < 0.01$), T4 group ($p < 0.05$) and the T5 group ($p < 0.01$) when compared with the control group. Defecation frequency was not significant between the negative control 2 group and the control group (not shown here).

## Hole-board test

In the hole-board test two parameters were observed, such as: line crossing, and head dipping.

Line crossing: In case of hole-board line crossing test, no difference was observed between the negative control 1 and the control group. When compared with the negative control 1 group, the T2, T3 group ($p < 0.01$), and the T4, T5 group ($p < 0.001$) showed highly significant differences i.e. lower line crossing. However, a significant difference was also observed in the T4 group with the control group ($p < 0.01$). In contrast, The T1 group did not show any significant difference over the negative control 1 group. Moreover, no significant differences were observed between the control group and the negative control 2 group (Fig 4A and 4B).

Head dipping: No significant difference was observed between the control and the negative control 1 group in the hole-board head dipping test. Significant differences were observed in the T3 ($p < 0.01$), T4 and T5 groups ($p < 0.001$) when compared with the negative control 1 group. However, the T3 ($p < 0.01$), T4, and the T5 groups also showed significantly lower number of head dipping in comparison with the control group ($p < 0.001$) (Fig 4C and 4D). No significant differences were observed between the control group and the negative control 2 group (not shown in the figure).

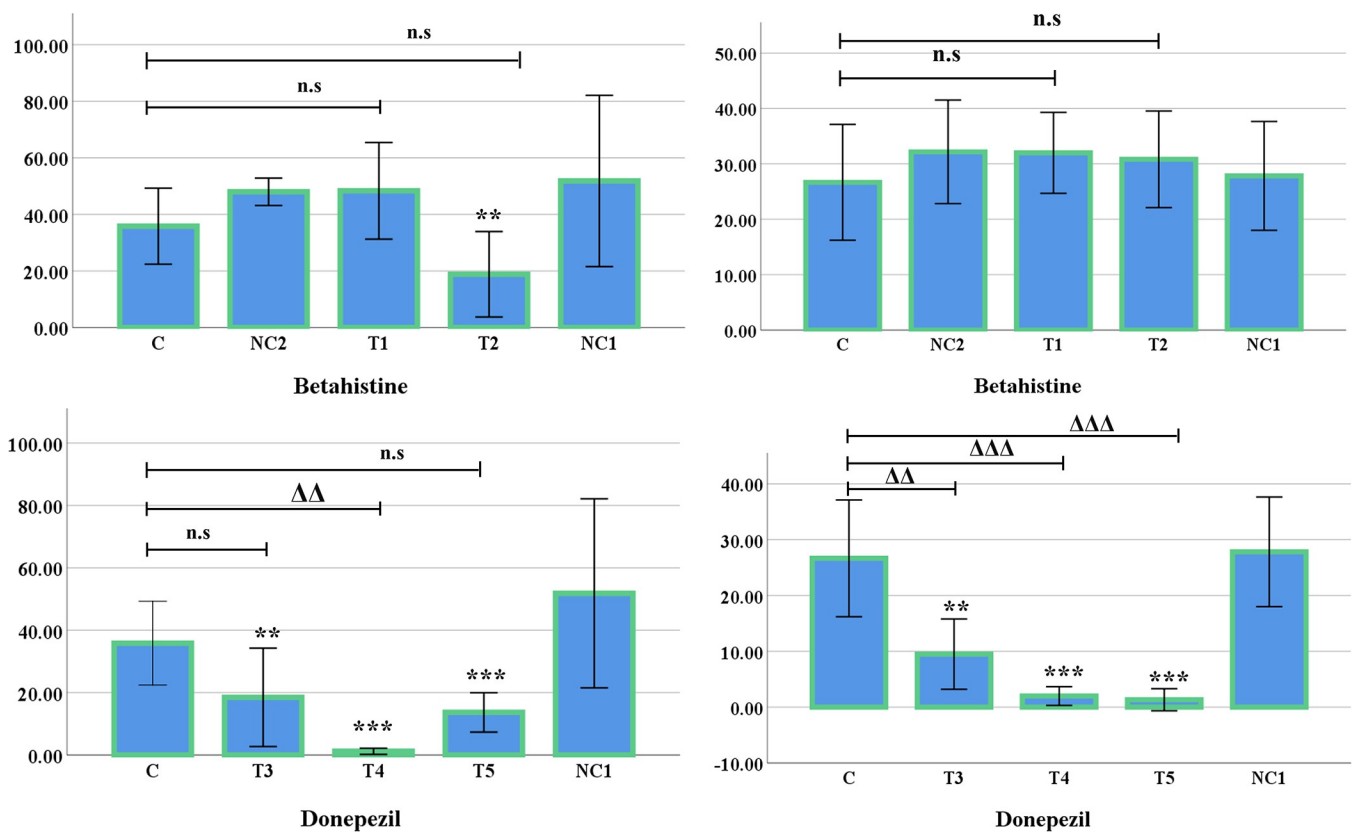

**Fig 4. Graphical representation of Hole-board test (Mean±SE).** (A&B) Line cross test of Betahistine and Donepezil, respecgively. (C&D) Head Dipping test of Betahistine and Donepezil, respectively. The following symbols: *, **, *** and Δ, ΔΔ, ΔΔΔ represents level of significance (*p*-values) at 5%, 1% and 0.1% level, when compared the treated groups with negative control 1 group and control group, respectively; n.s indicates non-significant.

## Hole-cross test

In the hole-cross test, a significant increase was observed in the negative control 1 group when compared with the control group ($p <0.001$). The T2 group showed significant decline over the negative control 1 group ($p <0.01$). Moreover, the T3, T4 and the T5 group showed significant decline ($p <0.001$) in hole crossing when compared with the negative control 1 group. However, the T1 group did not show any significant difference over the negative control 1 group. Furthermore, the T1 group had a significant difference with the control group ($p <0.001$) (Fig 5A and 5B). No significant differences were observed between the control group and the negative control 2 group (not shown in the figure).

## Histopathology

Microscopic observation (qualitative) of the control group and the negative control 2 group were unremarkable. In negative control 1 group, neuronal loss in cornue ammonis and subculum area and a few degenerated neurons (both dark and eosinophilic) were observed. Treated group 1 showed mild gliosis, neuronal loss in cornue ammonis, and dilated ventricles. In treated group 2, neuronal degeneration, a few apoptotic cells, neuropil vacuation and marked neuronal loss in cerebral cortex and hippocampus were observed. Treated group 3 showed thinning of hippocampus, degenerated neuron and moderate gliosis in cerebral cortex. In treated group 4, degenerated neurons in cerebral cortex and hippocampus, thinning of

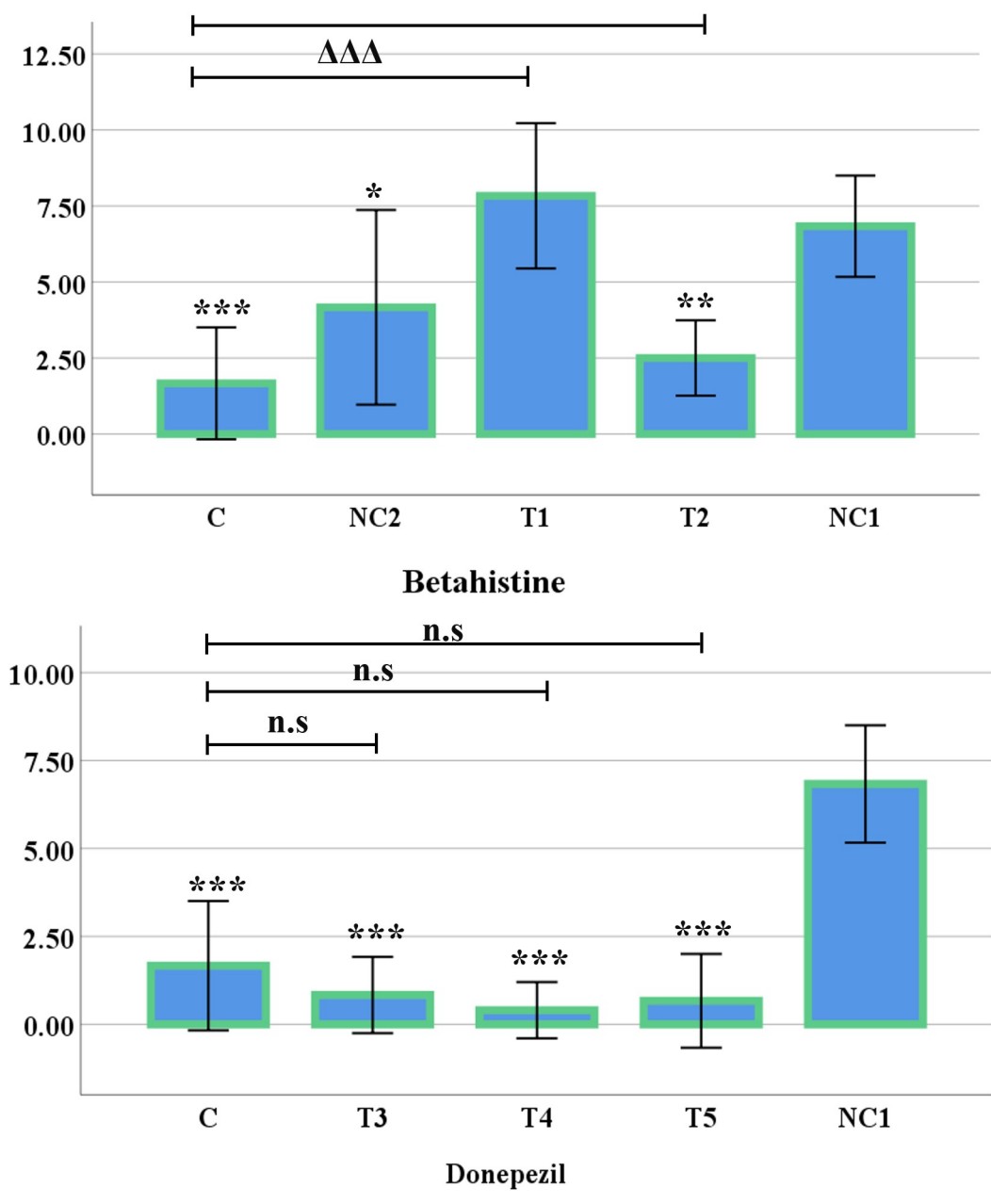

**Fig 5. Graphical representation of Hole-cross test (Mean±SE).** (A&B) Hole-cross test of Betahistine and Donepezil, respectively. The following symbols: *, **, *** and Δ, ΔΔ, ΔΔΔ represents level of significance (*p*-values) at 5%, 1% and 0.1% level, when compared the treated groups with negative control 1 group and control group, respectively. n.s indicates non-significant.

hippocampus, moderate gliosis in the cerebral cortex were observed. In the treated group 5, degenerated neurons in amygdala and neuronal loss in cortex were observed. Negative control 1 group showed more neuronal losses in cerebral cortex and hippocampus than the treated groups of mice. Moreover, reactive gliosis were more in the treated groups (Fig 6A-6H).

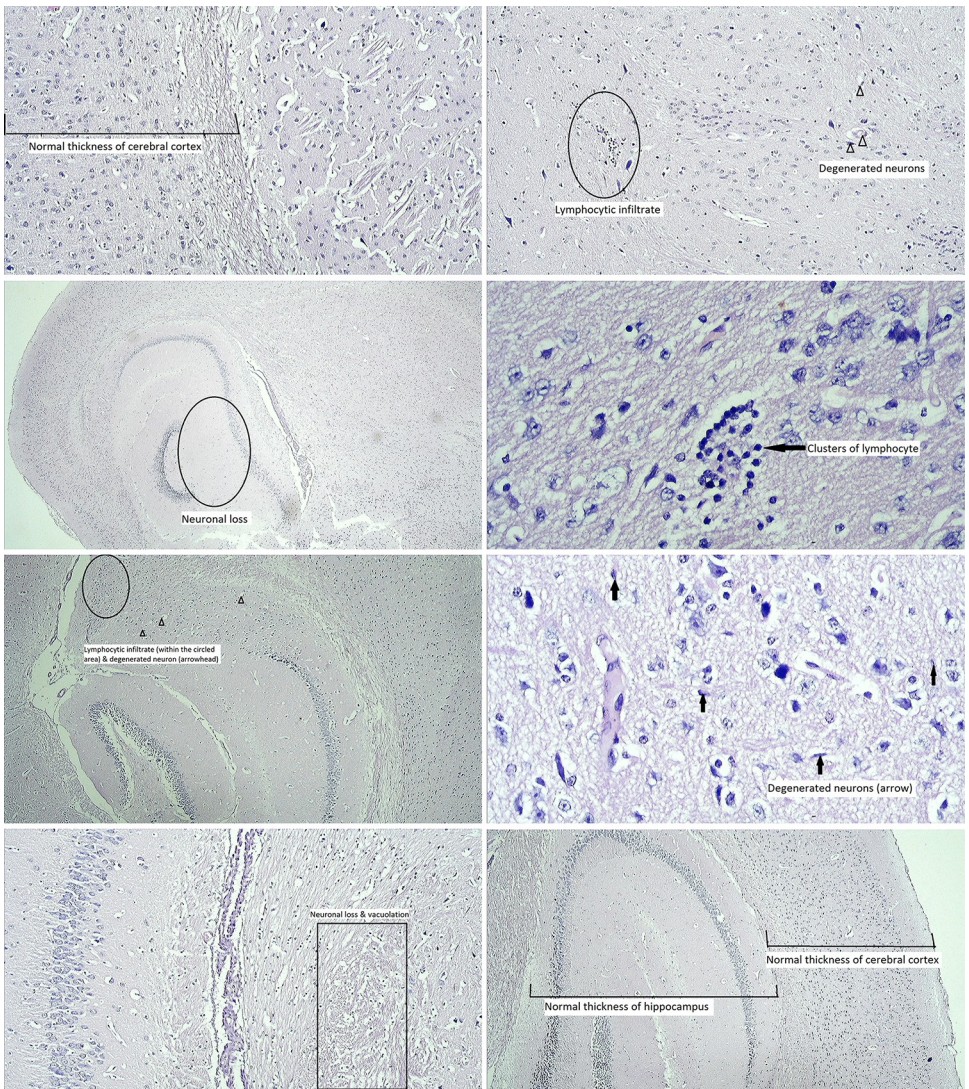

**Fig 6. Histopathology of the mouse brain.** (A) Unremarkable microscopic observation in the control group (100X). (B) Neuron degeneration was observed in the negative control 1 group (40X). (C) Mild gliosis, neuronal loss in cornue ammonis, and dilated ventricles were observed in treated group 1 (40X). (D) Neuronal degeneration, a few apoptotic cells, and neuronal loss were observed in Treated group 2 (100X). (E) Thinning of hippocampus, degenerated neuron and moderate gliosis were observed in treated group 3 (100X). (F) Thinning of hippocampus, degenerated neuron and moderate gliosis were observed in treated group 4 (400X). (G) Degenerated neurons in amygdala and neuronal loss in the cortex were observed in treated group 5 (400X). (H) Unremarkable microscopic observation in negative control 2 (40X).

## Discussion

This study examined the behavioural and brain-histopathological effect of Betahistine and Donepezil on VPA-exposed mice. Betahistine is an H3R antagonist and Donepezil is an acetyl-cholinesterase inhibitor. These two drugs with different doses evaluated short-term memory function, locomotor activities, and fear, anxiety-related behaviour of autistic *Swiss albino* mice. The Y-maze test in our experiment showed that these two drugs may play an important role in the retention of short-term memory function. Significantly increased locomotor activities in the negative control 1 group compared to the control group proves the hyperactivity of the

VPA-exposed. By performing different behavioural tests, Betahistine and Donepezil showed reduced hyperactivity at different doses. Moreover, negative control 1 group showed more neuronal losses in cerebral cortex and hippocampus than the treated groups of mice.

The T3 group shows similar behavioural outcomes as the control group in amelioration of hyperactivity, and anxiety, and preserving short-term memory in our tests and in other studies as well [32]. It is noteworthy that, in fear-induced response tests for novel stimuli hole board head dipping, Fig 5B there was no significant difference between the healthy (control) and the ASD-inflicted mice (negative control group 1). Alternatively, the T1 and the T2 groups showed promising results in ameliorating anxiety-induced hyperactivity and short-term memory preservation, respectively. The effects of the T1 and the T2 dose are mutually exclusive, whereby one does not influence the effect of the other.

Induction of VPA during the neural tube closure period showed higher occurrence of development of autism [19,24]. Since nervous system formation occurs during the embryonic days 11-17, a high dose of valproic acid (600 mg/kg) was injected intraperitoneally into pregnant mother mice on the embryonic 12.5th day to ensure the development of autism in offspring [23,25,26,33,34]. VPA is reported to change the physiology of the progenitor cells in the developing foetus in such a way that they tend to up-regulate acetylcholinesterase during the foetal development period while persisting even after birth, possibly throughout life [19,24]. Interestingly, alterations in the excitatory and inhibitory balance lie behind various traits of ASD. Elevation of E/I ratio (Alterations in the balance between neuronal excitation and inhibition) in the medial prefrontal cortex or mice introduced ASD-like symptoms in them which serve as an indicator of increased excitatory signals to be a key mediator behind their impaired behavioural effects [5].

In this study Betahistine, an H3R antagonist was studied to observe for behavioural improvement in ASD. Previously different doses of Betahistine for mice, for example 0.3 mg/kg, 3 mg/kg, 30 mg/kg and 2 mg/kg of body weight were used [35]. Betahistine dose of 0.3 mg/kg is recognized as suboptimal for H3R inhibition. In addition, 2 mg/kg oral dose of Betahistine for 8 days creates drug tolerance and undergoes first-pass metabolism [36]. Henceforth, these doses were abandoned for this study, while 30 mg/kg of Betahistine and 3 mg/kg of Betahistine along with 0.8 mg/kg of Rasagiline were included [35,37,38].

The oral dose of 0.3 mg/kg, 0.6mg/kg and 1 mg/kg of Donepezil was selected for this experiment based on previous reporting [32]. Acetylcholine gives its effects upon binding with acetylcholine receptors namely nicotinic and muscarinic receptors. However, for Donepezil, it is extremely hard to pinpoint which receptors give the desired effect [39]. Moreover, whether agonists or antagonists should be used also remains an issue. For this particular reason acetylcholinesterase inhibitors were used to inhibit the breakdown of already existing choline analogues.

The Y-maze spontaneous alternation test as a viable indicator to evaluate short-term memory of the test subjects presented significant decline ($p < 0.001$) of spontaneous alternation % in the negative control 1 group in comparison with the control group. This suggests impaired short-term memory which is a trait of ASD phenotypes [40]. In contrast, a significant development in preserving short-term memory was observed in the T1, T3, T4, and T5 groups.

The increased spontaneous percent alternation in the T1 group suggests amelioration of short-term memory deficits though the mechanism is not clear. The H3R is a G-protein coupled receptor which functions as an autoreceptor, controlling the release of histamine as well as a heteroreceptor, modulating release of neurotransmitters [41]. The development in T1 group might be due to changes caused by H3R heteroreceptor antagonism. It has been noted that Betahistine along with Rasagiline, brings forth a change in the pattern of release of neurotransmitters such as dopamine, serotonin, acetylcholine and GABA, by histaminergic neurons

in the tuberomammillary nucleus in the posterior hypothalamus region of the brain [2,15,42]. It is noteworthy that H1R loss of function can bring memory enhancing effects [15]. Betahistine, being a partial agonist of H1R [14], improved the memory deficit conditions in the T1 group which may be likely due to competitive binding with full agonists. Interestingly, the T2 group presented less significant development. Although the Betahistine dose was higher, MAO-B inhibitors were not used in this case. Previously, Betahistine has been regarded as a modulating neurotransmitter for histaminergic system due to its affinity for H1 and H3 receptors which is comparable to histamine while presenting no affinity towards H2 receptors [43]. Saturation of the H3 or H1 receptors might be responsible in this regard since the MAO-B was not available here to reduce the oxidation of the neurotransmitters.

Histamine level in the brain is low but its turnover is high [15]. Hence the less availability of histamine might be responsible for the change through partial H1R agonism due to the competitive binding with full agonists. Moreover, H3 antagonists can decrease neurodegeneration by increasing phosphorylation of intracellular proteins which are important to carry out the neurodegeneration process [42]. Studies have reported that 0.8mg/kg of Rasagiline alone does not have significant neuroprotective effects in transgenic mice models of neurodegeneration [37]. This finding strongly implies that the enhanced effect found in this result is due to the synergistic effects of Betahistine and Rasagiline used in combination and not due to neuroprotective abilities of MAO-B inhibitor Rasagiline only. H3R antagonists can be potential therapeutic agents for treating memory impairments and hyperactivity as H3R were previously targeted for Alzheimer's disease and attention deficit hyperactivity disorder (ADHD) treatment [22,41,44,45]. In the study, evidence in support of the claim was observed in the Negative Control 2 group (NC2) with no significant change in short-term memory retention when compared with the Control group (C).

Inhibition of breakdown of acetylcholine analogues might be a key factor in T3, T4 and T5 groups' development in preserving short term memory. However, the increment of dose does not show a linear relationship with the behavioural outcomes. The reason for this phenomenon is unclear. The cholinergic system has been reported to regulate memory, attention and cognitive flexibility, it is possible that the shortage of adequate acetylcholine due to the rightward shift in the upregulation of acetylcholinesterase or choline deficiency has been corrected or ameliorated to some extent by the use of Donepezil. A trend of downregulation of M1 muscarinic receptors and several nicotinic receptors have been reported in ASD phenotypes with an increase in 7 subunit nicotinic receptors [22,39,41,46]. Without establishing their function in the nervous system, it is very hard to theorize their part in the pathophysiology of ASD although their role in ASD is almost certain [46].

The significantly increased line crossing and rearing in the negative control group 1 in comparison with the control group suggests the development of hyperactivity which can also be observed in ASD phenotypes. A significant decrease of line crossing can be observed in the T2, T3, T4 and T5 group which is suggestive of mitigation of hyperactivity. However, the T5 group has decreased horizontal movement severely beyond desired level as the effect is significantly lower than the control group.

Nevertheless, the mice in Donepezil groups (T3, T4, T5) were significantly less active in the Y-maze test although the alternations were accurate. This behaviour of significantly reduced thigmotaxis is also reflected in the Open Field, Hole Board and Hole Cross tests.

The vertical and horizontal movement in the apparatus is significantly reduced which is an indicator of anxiogenic effects. In the open field line crossing, rearing and central area crossing test, the significant decline in all the treated groups reflects the anxiogenic effect when compared with the untreated VPA-exposed mice (NC1). In other word, it can also be said that Betahistine and Donepezil may have the ability to reduce hyperactivity. Interestingly, the two

doses of Betahistine (T1 & T2) showed increased activities and Donepezil doses (T3, T4, & T5) showed significantly decreased activities when compared with the healthy mice (C). However, to conclude about the anxiolytic and anxiogenic effect of both the drugs, more studies needs to be conducted.

The open field defecation test is suggestive of development of anxiety-like traits in the negative control 1 group. Although the T3 group ameliorated the symptom, the frequency of defecation decreased below desired level.

The hole-board line crossing tests' inability to show significant difference between the control group and the negative control 1 group is suggestive of normal anxiogenic response under fear-based conditions in valproic acid-induced mice. That said, the T4 group and the T5 group have significantly decreased the line crossing.

Similarly, the absence of significant difference in head dipping frequency suggests that the neophilic-neophobic response was normal as well. The significant changes observed in the T3, T4 and T5 group is suggestive of anxiogenic activity, although further investigations are required to support these data.

A decline was observed in hole crossing frequency in the T2 group. Betahistine does not show sedative properties [47]. Nonetheless, H1R loss of function can lead to anxiolytic effects [14]. It is possible that the change was brought about by partial H1R agonism due to competitive binding with full agonists. Another probable reason could be histamine H1R tolerance since Betahistine can increase histamine synthesis by blocking H3R autoreceptors [33,48]. However, in another experiment conducted on similar specimen [36], 30 mg/kg of oral dose of Betahistine for 8 days shows significant increase in t-MeHA, (tele-methylhistamine) which is an index for measuring histaminergic neuronal activity. This data strongly suggests that inverse agonism in H3R autoreceptors result in the amplification of activities in the histaminergic neurons [36,41]. Therefore, the behavioural activities of T1 and T2 groups of mice found in our study could also result from the amplification of histaminergic neuronal activities due to H3R inverse agonism.

The decline in hole crossing frequency observed in the Donepezil treated groups (T3, T4, and the T5 group) for these tests were unlikely brought about by sedation since Donepezil has not yet been reported to cause sedative effects. However, in different studies it has been shown that, choline concentrations are negatively associated with anxiety symptoms [41,49,50]. It is possible that the increase in acetylcholine might be the reason behind the significantly decreased hole crossing in the Donepezil treated groups.

The mice in the NC1 group and the treated groups showed neuronal loss and degeneration, gliosis, and thinning of hippocampus. The hippocampus of the brain is responsible for the formation of new memory and the processing of short-term memory into long term memory [51]. Changes in brain-histomorphology were prominent in the NC1 group compared to treated groups of mice. However, no measurement has been conducted to quantify the difference. Due to the lack of neuronal regeneration in the brain, the treated groups of mice presented neuronal degeneration. In addition to it, reactive gliosis, where more glial cells supporting the nerve cells are created, was observed more in the treated groups. However, the precise origin and subsequent fate of the glial cells reacting to injury are unknown [52].

## Conclusion

This study was conducted to evaluate if Betahistine or Donepezil can ameliorate the symptoms of autism. The results suggest both the drugs have potential to ameliorate short-term memory deficits. Furthermore, all the doses of the two drugs have the potential to reduce hyperactivity. Further research should be continued for the proper justification of the effects of the drugs.

Therefore, both Betahistine (alone and in combination with Rasagiline) and Donepezil (low dose 0.3mg/kg) have potential as a therapeutic agent to ameliorate some of the symptoms found in ASD affected individuals. It can play a role in managing hyperactivity in individuals with ADHD, however, more study will be required to understand its mechanism of action as well as studies into its toxicity effects.

## Supporting information

**S1 Dataset. Dataset of behavioural traits of valproic acid-induced mouse model.** (PDF)

## Acknowledgments

We are thankful to Sanofi Aventis Bangladesh, Square Pharmaceuticals Limited, Bangladesh, for providing the chemicals used in this study. We also acknowledge our gratitude to the Pharmacology Laboratory at Jahangirnagar University, for providing us laboratory support. Lastly, but not the least, we also acknowledge the contribution of Fariha Tarannum in referencing.

## Author Contributions

**Conceptualization:** Sharmind Neelotpol, Rifat Rezwan, Timothy Singh.

**Data curation:** Sharmind Neelotpol, Rifat Rezwan, Timothy Singh, Iffat Islam Mayesha, Sayedatus Saba, Mohd Raeed Jamiruddin.

**Formal analysis:** Sharmind Neelotpol, Iffat Islam Mayesha, Sayedatus Saba, Mohd Raeed Jamiruddin.

**Investigation:** Sharmind Neelotpol, Rifat Rezwan, Timothy Singh, Iffat Islam Mayesha, Sayedatus Saba.

**Methodology:** Sharmind Neelotpol, Rifat Rezwan, Timothy Singh, Mohd Raeed Jamiruddin.

**Project administration:** Rifat Rezwan, Timothy Singh.

**Resources:** Rifat Rezwan, Timothy Singh.

**Supervision:** Sharmind Neelotpol.

**Validation:** Sharmind Neelotpol, Mohd Raeed Jamiruddin.

**Writing – original draft:** Sharmind Neelotpol, Rifat Rezwan, Timothy Singh, Iffat Islam Mayesha, Mohd Raeed Jamiruddin.

**Writing – review & editing:** Sharmind Neelotpol, Rifat Rezwan, Timothy Singh, Iffat Islam Mayesha, Sayedatus Saba, Mohd Raeed Jamiruddin.

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
