## [Decision Letter · Decision Letter 0]

10 Nov 2023

PONE-D-23-30357Pharmacological intervention of behavioural traits of autism spectrum disorder in a prenatal valproic acid-induced mouse model of autismPLOS ONE

Dear Dr. Neelotpol,

Thank you for submitting your manuscript to PLOS ONE. After careful consideration, we feel that it has merit but does not fully meet PLOS ONE’s publication criteria as it currently stands. Therefore, we invite you to submit a revised version of the manuscript that addresses the points raised during the review process.

We look forward to receiving your revised manuscript.

Kind regards,

Tommaso Martino, M.D.

Academic Editor

PLOS ONE

2. We note that Figure 2 in your submission contain copyrighted images. All PLOS content is published under the Creative Commons Attribution License (CC BY 4.0), which means that the manuscript, images, and Supporting Information files will be freely available online, and any third party is permitted to access, download, copy, distribute, and use these materials in any way, even commercially, with proper attribution. For more information, see our copyright guidelines: http://journals.plos.org/plosone/s/licenses-and-copyright.

1. You may seek permission from the original copyright holder of Figure 2 to publish the content specifically under the CC BY 4.0 license.

Reviewers' comments:

Reviewer's Responses to Questions

**Comments to the Author**

1. Is the manuscript technically sound, and do the data support the conclusions?

Reviewer #1: Yes

Reviewer #2: Partly

Reviewer #3: Partly

2. Has the statistical analysis been performed appropriately and rigorously? 

Reviewer #1: Yes

Reviewer #2: Yes

Reviewer #3: I Don't Know

3. Have the authors made all data underlying the findings in their manuscript fully available?

Reviewer #1: Yes

Reviewer #2: Yes

Reviewer #3: No

4. Is the manuscript presented in an intelligible fashion and written in standard English?

Reviewer #1: Yes

Reviewer #2: Yes

Reviewer #3: No

5. Review Comments to the Author

Reviewer #1: - A Figure is needed to show the affinities and selectivity of test compound betahistine.

- When discussing the involvement of histaminergic and/or cholinergic neurotransmission in the observed effects for betahistine, cite the following siginificant studies in your discussion;

1) Eissa N, Al Awad M, Venkatachalam K, Jayaprakash P, Thomas SD, Zhong S, Stark H, Sadek B. Simultaneous antagonism at H3R/D2R/D3R reduces autism-like self-grooming and aggressive behaviors by mitigating MAPK activation in mice. Int. J. Mol. Sci. 2023; 24:526.

2) Saad AK, Akour A, Mahboob A, AbuRuz S, Sadek B. Role of brain modulators in neurodevelopment: focus on autism spectrum disorder and associated comorbidities. Pharmaceuticals 2022, 15:612.

3) Eissa N, Venkatachalam K, Jayaprakash P, Falkenstein M, Frank A, Reiner-Link D, Stark H, Sadek B. The multi-targeting ligand ST-2223 with histamine H3 receptor and dopamine D2/D3 receptor antagonist properties mitigates autism-like repetitive behaviors and brain oxidative stress in mice. Int. J. Mol. Sci. 2021; 22(4), 1947.

4) Eissa N, Azimullah S, Jayaprakash P, Jayaraj RL, Reiner D, Ojha SK, Beiram R, Stark H, Łażewska D, Kieć-Kononowicz K, Sadek B. The dual-active histamine H3 receptor antagonist and acetylcholine esterase inhibitor E100 alleviates autistic-like behaviors and oxidative stress in valproic acid induced autism in mice. Int. J. Mol. Sci. 2020; 21(11), 3996.

- References to justify the selected and used doses of all test compounds are missing.

-Quality of all Figures should be improved, were difficult to read.

Reviewer #2: 1. The authors chose to treat the rats from PND44 to 64. Why this particular time window, frequency and duration of treatment? What is the authors' hypothesis when treating after weaning, when the neurodevelopmental period is mostly completed? Of note, relieving effects of treatment observed in nearly young adults are encouraging.

2. The term ‘Alteration’ and ‘alternation’ is written interchangeably in Y maze in methods and results section. It should be spontaneous alternation and not alteration. Correct it.

3. Data representation: histograms with scatter plots are highly recommended to allow the reader to assess the animal numbers per group and the dispersion of individual data at a glance.

4. Autistic mice: avoid using such wording; use the terms "VPA-exposed mice" instead.

5. In abstract it is mentioned that “After administration of the experimental doses, various locomotor tests: Open Field, Hole-Board, Hole Cross tests and behavioural tests: Y-Maze Spontaneous Alternation Test were performed and compared with positive and negative control groups”. What is the positive control group that is mentioned in this statement, which is nowhere mentioned in the rest of manuscript?

6. What is the significance of combining betahistine and rasagiline, why were both the drugs given in combination in T1 group? Kindly add the significance in the manuscript.

7. Authors have concluded that drugs have potential to ameliorate short-term memory deficits, it is recommended to perform other behavioral parameters for memory assessment like, Morris water maze, Radial Arm maze and Novel object recognition, to have better glance into the cognitive outcomes.

8. Furthermore it is recommended to report the histopathological changes in the appropriate regions of the brain of VPA exposed mice, using hematoxylin and eosin stain and Nissl stain, and also estimation of levels of acetyl choline and histamine in the VPA exposed mice and treated mice is highly recommended. These basic investigations will make the preliminary findings from this study more reliable and convincing for further research into these molecules and their mechanisms in ASD.

Reviewer #3: 1. In Lines 23 and 26 leave space after ‘p’ to be consistent. Apply this to the results section too.

2. Sometimes the authors mentioned autism and other times ASD. Please be consistent.

3. Why were the 2 mice genders used? Will this not have any impact on the behavior results?

4. I suggest that the second group of healthy offspring from the healthy mothers (NC2) to be Betahistine dihydrochloride of 3mg/kg in combination with 0.8mg/kg of Rasagiline to ensure that there are no off-target effects due to the combination of the drugs. This is important as especially there is no absence of in vitro profile of the combination of these drugs. Why the current NC2 is used?

5. Is there any results for the effects of Rasagiline only in VPA-induced mouse of ASD on all behavioral tests studied?

6. Line 343 mentions ‘In the study evidence in support of the claim was observed in the Negative control 2 group (NC2) whereby no significant change in short-term memory retention when compared with the Control group (C). To test for the neuroprotective effect of rasagiline, the effect of rasagiline only should be tested on an ASD mouse model of ASD. This group is missed.

7. The quality of all the figures is poor. Figures with high resolution need to be added.

8. In Figure 1, add the full name of the behavioral tests or mention them in the figure caption.

9. In Line 223, 224, and 343 the control and negative control 1 should all be in small letters as it is throughout the paper.

10. Elaboration is required on how betahistine showed anxiolytic and donepezil showed anxiogenic properties at different doses.

11. In Line 286 remove the bracket.

12. In Lines 292 and 292 ‘embryonic’ should be in small letters.

13. In 294 remove the extra space.

14. The authors should mention VPA-induced mice, not valproic acid-induced mice throughout the paper. The abbreviations should be described the first time in the text only.

15. In line 404, rephrase the sentence. There are no types of ASD patients.

16. Please check the manuscript for the English language.

6. PLOS authors have the option to publish the peer review history of their article (what does this mean?). If published, this will include your full peer review and any attached files.

Reviewer #1: **Yes: **Bassem Sadek

Reviewer #2: No

Reviewer #3: **Yes: **Nermin Eissa

---

## [Author Response · Author response to Decision Letter 0]

20 Apr 2024

Response: We have made sure all the PLOS ONE’S style requirements for fie naming have been followed throughout the manuscript. 

2. We note that Figure 2 in your submission contains copyrighted images. All PLOS content is published under the Creative Commons Attribution License (CC BY 4.0), which means that the manuscript, images, and Supporting Information files will be freely available online, and any third party is permitted to access, download, copy, distribute, and use these materials in any way, even commercially, with proper attribution. For more information, see our copyright guidelines: http://journals.plos.org/plosone/s/licenses-and-copyright.

1. You may seek permission from the original copyright holder of Figure 2 to publish the content specifically under the CC BY 4.0 license.

Response: Since we did not incorporate figures for any test, therefore, we have removed Figure 2 to avoid copyrighting issues. 

Response to the reviewers' comments:

Reviewer #1: 

● A Figure is needed to show the affinities and selectivity of test compound betahistine.

Response: Thank you for your suggestion. This study does not focus on the molecular mechanism of betahistine. Our aim was to study the behavioural mechanism of betahistine on VPA-induced mice. The affinity and selectivity of betahistine has already been established and information regarding its binding affinity and selectivity has been referred to in the revised manuscript in line 397-399 and line 401-403 with reference 43. On our next project we are considering studying the molecular mechanism of betahistine over its behavioural pattern. 

● When discussing the involvement of histaminergic and/or cholinergic neurotransmission in the observed effects for betahistine, cite the following significant studies in your discussion;

1) Eissa N, Al Awad M, Venkatachalam K, Jayaprakash P, Thomas SD, Zhong S, Stark H, Sadek B. Simultaneous antagonism at H3R/D2R/D3R reduces autism-like self-grooming and aggressive behaviours by mitigating MAPK activation in mice. Int. J. Mol. Sci. 2023; 24:526.

2) Saad AK, Akour A, Mahboob A, AbuRuz S, Sadek B. Role of brain modulators in neurodevelopment: focus on autism spectrum disorder and associated comorbidities. Pharmaceuticals 2022, 15:612.

3) Eissa N, Venkatachalam K, Jayaprakash P, Falkenstein M, Frank A, Reiner-Link D, Stark H, Sadek B. The multi-targeting ligand ST-2223 with histamine H3 receptor and dopamine D2/D3 receptor antagonist properties mitigates autism-like repetitive behaviours and brain oxidative stress in mice. Int. J. Mol. Sci. 2021; 22(4), 1947.

4) Eissa N, Azimullah S, Jayaprakash P, Jayaraj RL, Reiner D, Ojha SK, Beiram R, Stark H, Łażewska D, Kieć-Kononowicz K, Sadek B. The dual-active histamine H3 receptor antagonist and acetylcholine esterase inhibitor E100 alleviates autistic-like behaviours and oxidative stress in valproic acid induced autism in mice. Int. J. Mol. Sci. 2020; 21(11), 3996.

Response: Thank you for your suggestion. We have cited the aforementioned studies in our discussion. 

Reference 01 can be found as reference 41 in line number: 389,413, 423, 460, and 465. 

Reference 02 can be found as reference 46 in line number: 423 and 425.

Reference 03 can be found as reference 14 in line number: 68, 88, 395, and 453.

Reference 04 can be found as reference 24 in line number: 113, 124, and 358.

● References to justify the selected and used doses of all test compounds are missing.

Response: Thank you for your comment. The doses were reviewed from the referenced articles. Some doses were modified, which are justified in the discussion section. Betahistine and Rasagiline doses were justified in line 457-460 using reference no 33,36, 41 & 48 in the manuscript. Donepezil doses were justified in line 462-466 using reference no 41,49 & 50 in the manuscript. 

● Quality of all Figures should be improved, as they are difficult to read.

Response: Thank you for your suggestion. The quality of all the figures have now been improved. 

Reviewer #2: 

1. The authors chose to treat the rats from PND44 to 64. Why this particular time window, frequency and duration of treatment? What is the authors' hypothesis when treating after weaning, when the neurodevelopmental period is mostly completed? Of note, relieving effects of treatment observed in nearly young adults are encouraging.

Response: Your query is appreciated. The time window selected for dosage and performing of behavioural studies were done according to the referenced article number 23 of our manuscript. 

The dosages are administered after weaning and not at a young age.

2. The term ‘Alteration’ and ‘alternation’ is written interchangeably in Y maze in the methods and results section. It should be spontaneous alternation and not alteration. Correct it.

Response: Thank you for your notification. Recommended changes have been made accordingly in the line numbers 216 & 432. 

3. Data representation: histograms with scatter plots are highly recommended to allow the reader to assess the animal numbers per group and the dispersion of individual data at a glance.

Response: We appreciate this valuable suggestion. In this manuscript, the quality of the existing figures have been improved according to the other two reviewer’s suggestion. In addition, as per your suggestion, the number of animals per group have been mentioned in the method section.

4. Autistic mice: avoid using such wording; use the terms "VPA-exposed mice" instead.

Response: Thank you for your notification. Changes have been made accordingly. (Line-26,33, 64.82,342,347,437)

5. In abstract it is mentioned that “After administration of the experimental doses, various locomotor tests: Open Field, Hole-Board, Hole Cross tests and behavioural tests: Y-Maze Spontaneous Alternation Test were performed and compared with positive and negative control groups”. What is the positive control group that is mentioned in this statement, which is nowhere mentioned in the rest of the manuscript?

Response: Thank you for your reply. In Line 23 the control group is mentioned as 'positive control'. The word has been changed to control instead of positive control.

6. What is the significance of combining betahistine and rasagiline, why were both the drugs given in combination in the T1 group? Kindly add the significance in the manuscript.

Response: Thank you for your query. The significance of combining betahistine and rasagiline has been mentioned in line 406-411.

7. Authors have concluded that drugs have potential to ameliorate short-term memory deficits, it is recommended to perform other behavioural parameters for memory assessment like, Morris water maze, Radial Arm maze and Novel object recognition, to have a better glance into the cognitive outcomes.

Response: We are grateful for the suggestion you have provided. We did not mention about these tests in the ethical permission, hence it is not possible for us to conduct these experiments. Moreover, we have already sacrificed the mice for histopathological data. We will consider your valuable suggestions for future studies.

8. Furthermore it is recommended to report the histopathological changes in the appropriate regions of the brain of VPA-exposed mice, using hematoxylin and eosin stain and Nissl stain, and also estimation of levels of acetylcholine and histamine in the VPA-exposed mice and treated mice is highly recommended. These basic investigations will make the preliminary findings from this study more reliable and convincing for further research into these molecules and their mechanisms in ASD.

Response: Thank you for this important suggestion. Now we have incorporated the findings of histopathology which, we believe, have improved the quality of the study. 

Reviewer #3: 

1. In Lines 23 and 26 leave space after ‘p’ to be consistent. Apply this to the results section too.

Response: Thank you for your notification. Recommended changes have been made accordingly in line 24 and 27. For consistency, we have checked the whole manuscript carefully.

2. Sometimes the authors mentioned autism and other times ASD. Please be consistent.

Response: Thank you for your notification. Recommended changes have been made throughout the manuscript, accordingly. 

3. Why were the 2 mice genders used? Will this not have any impact on the behavior results?

Response: Thank you for your valuable comment. There was a shortage of VPA-induced mice pups to conduct the whole experiment with only one specific gender since some of the VPA- induced mother mice died before giving birth. Hence, we have compared the male treated groups with male control groups. Similarly, the female treated groups were compared with female control groups. We did not draw a contrast between the male or female groups.

4. I suggest that the second group of healthy offspring from the healthy mothers (NC2) to be Betahistine dihydrochloride of 3mg/kg in combination with 0.8mg/kg of Rasagiline to ensure that there are no off-target effects due to the combination of the drugs. This is important as especially there is no absence of in vitro profile of the combination of these drugs. Why is the current NC2 used?

Response: We extend our gratitude for your input. Both Rasagiline and Betahistine have passed clinical trials and their safety is not in question. Moreover, due to the shortage of mice pups this group was avoided. 

To negate the effects of rasagiline on VPA-induced mice we have used 0.8mg/kg of rasagiline on healthy mice as the negative control 2 group (NC2). The NC2 group helps us to understand if any derogatory effects arose from the administration of Rasagiline. 

5. Is there any results for the effects of Rasagiline only in VPA-induced mouse of ASD on all behavioral tests studied?

Response: We thank you for your query. We did not test the effects of 0.8mg/kg of Rasagiline only in VPA-induced mice. Sufficient VPA-injected mother mice did not survive to give birth to the required number of mice pups to make this VPA-induced group. An explanation of which has now been added in the methodology section. Moreover, the project did not aim to evaluate MAO-B Rasagiline’s effect separately on VPA induced mice. Furthermore, we have referenced a study that reported 0.8mg/kg of rasagiline alone could not significantly give any neuroprotective effects in transgenic mice models of neurodegeneration in line 409, reference 37.

6. Line 343 mentions ‘In the study evidence in support of the claim was observed in the Negative control 2 group (NC2) whereby no significant change in short-term memory retention when compared with the Control group (C). To test for the neuroprotective effect of rasagiline, the effect of rasagiline only should be tested on an ASD mouse model of ASD. This group is missed.

Response: Thank you for your suggestion. Rasagiline of 0.8mg/kg was added to T1 group to counteract the tolerance initiated by 3mg/kg of Betahistine in subjects after 8 days of administration, which has been discussed in the manuscript line 389-393 (Reference 2, 15, 42). The experiment did not aim to evaluate MAO-B inhibitor rasagiline’s effect on VPA induced mice. Hence this group was avoided. We will consider it for future studies. 

7. The quality of all the figures is poor. Figures with high resolution need to be added.

Response: Thank you for your suggestion. The quality of the figures have been improved now as per the suggestion. 

8. In Figure 1, add the full name of the behavioral tests or mention them in the figure caption.

Response: Thank you for your notification. We have added them accordingly. 

9. In Line 223, 224, and 343 the control and negative control 1 should all be in small letters as it is throughout the paper.

Response: Thank you for your notification. We have changed it throughout the manuscript.

10. Elaboration is required on how betahistine showed anxiolytic and donepezil showed anxiogenic properties at different doses.

Response: Thank you for your query. Since the focus of the study was to evaluate the behavioural trait of two drugs on VPA-induced mouse model, it is hard to make any comment about the anxiolytic and anxiogenic effect of the drugs by conducting only few neuro-behavioural tests. That is why, we have rephrased the sentence to better reflect our thought with regards to the experiments that we have carried out.

11. In Line 286 remove the bracket.

Response: Thank you for your comment. Bracket has been removed as per the advice.

12. In Lines 292 and 292 ‘embryonic’ should be in small letters. 

Response: Thank you for your comment. Recommended changes have been made accordingly. 

13. In 294 remove the extra space.

Response: Thank you for your notification. Extra space has been removed.

14. The authors should mention VPA-induced mice, not valproic acid-induced mice throughout the paper. The abbreviations should be described the first time in the text only.

Response: Thank you for your suggestion. We have changed it throughout the manuscript.

15. In line 404, rephrase the sentence. There are no types of ASD patients.

Response: Thank you for your insight. We have changed accordingly.

16. Please check the manuscript for the English language.

Response: We have checked the manuscript for the English language.

---

## [Decision Letter · Decision Letter 1]

29 Jul 2024

Pharmacological intervention of behavioural traits and brain histopathology of prenatal valproic acid-induced mouse model of autism

PONE-D-23-30357R1

Dear Dr. Neelotpol,  

We’re pleased to inform you that your manuscript has been judged scientifically suitable for publication and will be formally accepted for publication once it meets all outstanding technical requirements.

Kind regards,

Xiaona Wang, Ph.D

Academic Editor

PLOS ONE

Additional Editor Comments (optional):

Reviewers' comments:

Reviewer's Responses to Questions

**Comments to the Author**

1. If the authors have adequately addressed your comments raised in a previous round of review and you feel that this manuscript is now acceptable for publication, you may indicate that here to bypass the “Comments to the Author” section, enter your conflict of interest statement in the “Confidential to Editor” section, and submit your "Accept" recommendation.

Reviewer #2: All comments have been addressed

2. Is the manuscript technically sound, and do the data support the conclusions?

Reviewer #2: Partly

3. Has the statistical analysis been performed appropriately and rigorously? 

Reviewer #2: Yes

4. Have the authors made all data underlying the findings in their manuscript fully available?

Reviewer #2: Yes

5. Is the manuscript presented in an intelligible fashion and written in standard English?

Reviewer #2: Yes

6. Review Comments to the Author

Reviewer #2: Authors have addressed all the queries and with the histopathological data, the data is more convincing. However, authors could have provided the histopathological data for each interventional group.

7. PLOS authors have the option to publish the peer review history of their article (what does this mean?). If published, this will include your full peer review and any attached files.

Reviewer #2: **Yes: **Kajal Rawat

---

## [Editor Report · Acceptance letter]

13 Sep 2024

PONE-D-23-30357R1 

PLOS ONE

Dear Dr. Neelotpol, 

I'm pleased to inform you that your manuscript has been deemed suitable for publication in PLOS ONE. Congratulations! Your manuscript is now being handed over to our production team.

Kind regards, 

on behalf of

Associate Professor Xiaona Wang 

Academic Editor

PLOS ONE